# WorldPack: Compressed Memory Improves Spatial Consistency in Video World Modeling

## Abstract

Video world models have attracted significant attention for their ability to produce high-fidelity future visual observations conditioned on past observations and navigation actions. Temporally- and spatially-consistent, long-term world modeling has been a long-standing problem, unresolved with even recent state-of-the-art models, due to the prohibitively expensive computational costs for long-context inputs. In this paper, we propose *WorldPack*, a video world model with efficient compressed memory, which significantly improves spatial consistency, fidelity, and quality in long-term generation despite much shorter context length. Our compressed memory consists of trajectory packing and memory retrieval; trajectory packing realizes high context efficiency, and memory retrieval maintains the consistency in rollouts and helps long-term generations that require spatial reasoning. Our performance is evaluated with LoopNav, a benchmark on Minecraft, specialized for the evaluation of long-term consistency, and we verify that WorldPack notably outperforms strong state-of-the-art models.

## 1 Introduction

Video world models, i.e., neural world simulators based on video generation models, have recently attracted significant attention for their ability to produce high-fidelity future visual observations conditioned on past observations and navigation actions (Brooks et al., 2024; Bruce et al., 2024; Kang et al., 2024). By predicting and generating future visual observations from past observations and agent actions, these models hold the potential to serve as alternatives to conventional simulation environments. Their applications span a wide range of domains, such as robotic simulation (Bar et al., 2024; Hu et al., 2025; Zhu et al., 2025), autonomous driving (Hu et al., 2023; Russell et al., 2025; Wang et al., 2023; Zhao et al., 2024; Gao et al., 2024), and AI-driven content generation in game engines (Alonso et al., 2024; Valevski et al., 2024).

Despite this promise, achieving temporally and spatially consistent world modeling over long horizons remains a long-standing challenge, even with recent state-of-the-art video generation models (Decart et al., 2024; Guo et al., 2025). This difficulty stems from the prohibitively high computational cost required to process long-context inputs, which limits existing models to relatively short temporal windows (Alonso et al., 2024; Bar et al., 2024). As a result, previously observed information is easily discarded, leading to inconsistencies in spatial layouts and object arrangements over time. For instance, an object visible in one view may abruptly vanish or shift position when the perspective changes, undermining the reliability of such models as world simulators.

In this paper, we propose *WorldPack*, a long-context-aware video world model that achieves efficient compressed memory while maintaining high generation quality. Despite operating with relatively short context lengths, WorldPack substantially improves long-term spatial consistency. The compressed memory consists of two key components: *trajectory packing*, which enhances context efficiency by retaining more recent information in a compact form, and *memory retrieval*, which selectively recalls past scenes that share substantial visual overlap with the prediction target. Together, these mechanisms ensure consistent rollouts even in later stages, where reliable spatial reasoning is crucial. We adopt conditional diffusion transformer (CDiT) (Bar et al., 2024) as a base backbone architecture and incorporate RoPE-based temporal embeddings (Su et al., 2023), enabling effective utilization of memories regardless of their temporal distance from a target scene.

Our experiments evaluate WorldPack on LoopNav (Lian et al., 2025), a benchmark designed to assess long-horizon temporal- and spatial-consistency in a Minecraft-based environment. On both the spatial memory retrieval task, which measures the ability to recall past observations, and the spatial reasoning task, which evaluates consistency under long-horizon rollouts, WorldPack demonstrates superior scene prediction performance. Notably, it substantially outperforms strong state-of-the-art baselines such as Oasis, MineWorld (Guo et al., 2025), Diamond (Alonso et al., 2024), and NWM (Bar et al., 2024), as validated across multiple quality metrics, including SSIM (Wang et al., 2004), LPIPS (Zhang et al., 2018), PSNR, and DreamSim (Fu et al., 2023).

## 2 RELATED WORK

**Video World Models.**   Recent advances in video diffusion models have enabled photorealistic, high-resolution video generation, positioning them as "general-purpose world simulator" capable of producing diverse scenes with plausible dynamics from text (Brooks et al., 2024; Google DeepMind, 2024; Kang et al., 2024; Bansal et al., 2024; Chefer et al., 2025; Wu et al., 2025; Oshima et al., 2025). Building on this progress, video world models have attracted significant attention for their ability to generate high-fidelity future visual observations conditioned on past scene sequences and navigation actions (Alonso et al., 2024; Bruce et al., 2024; Mao et al., 2025). Their applications span a wide range of domains, such as game engines (Valevski et al., 2024; Decart et al., 2024; Guo et al., 2025), autonomous driving (Hu et al., 2023; Russell et al., 2025; Wang et al., 2023; Zhao et al., 2024; Gao et al., 2024), (Hu et al., 2024; Guo et al., 2024), and robotics (Bar et al., 2024; Hu et al., 2025; Zhu et al., 2025). These studies underscore the importance of maintaining long-term temporal and spatial consistency, particularly in decision-making tasks such as driving and navigation. However, achieving such coherence remains an unresolved challenge, even for state-of-the-art models, due to the prohibitively high computational costs required to process a long sequence of observations in the model context (Decart et al., 2024; Guo et al., 2025).

**Long-Context Video Generation.**   Prior works in video generation have actively explored ways to extend fixed-length horizons into long-term rollouts. One line of research focuses on sampling strategies, such as temporal super-resolution with coarse-to-fine processing (Ho et al., 2022b; Yin et al., 2023), autoregressive generation conditioned on recent frames (He et al., 2022; Henschel et al., 2024), and inference-time techniques that adapt pretrained models for longer generations without retraining (Qiu et al., 2023; Kim et al., 2024). Another direction introduces architectural advances to capture long-range dependencies, including structured state space models (Gu et al., 2021; Gu & Dao, 2023) for efficient temporal modeling (Oshima et al., 2024; Po et al., 2025) and spatial retrieval mechanisms that dynamically select past frames with overlapping fields of view (Yu et al., 2025; Xiao et al., 2025). In parallel, stabilization methods mitigate degradation during long generations, for example, by combining next-token prediction with full-sequence diffusion (Chen et al., 2024; Ruhe et al., 2024; Jin et al., 2024; Kodaira et al., 2025) or by incorporating history-based guidance to preserve past information (Song et al., 2025). Recently, Zhang & Agrawala (2025) proposes to compress past frames at varying rates into the context to balance efficiency and long-term consistency. We transfer such a technique for long-context generation in the context of video world modeling, where preserving spatial coherence under action-conditioned rollouts poses distinct challenges in many downstream tasks (e.g., robotics, self-driving, etc), and demonstrate that compressing retrieved past states helps improve spatial reasoning in long-context rollouts.

## 3 PRELIMINARIES

We begin by extending latent diffusion models (Rombach et al., 2022) to the temporal domain, formulating video diffusion models (He et al., 2022; Ho et al., 2022a). Given a sequence of frames $\mathbf{x}_{0:T} = (\mathbf{x}_0, \mathbf{x}_1, \ldots, \mathbf{x}_T)$, we first encode frames into latent representations $\mathbf{z}_{0:T} = (\mathbf{z}_0, \mathbf{z}_1, \ldots, \mathbf{z}_T)$ using a pretrained VAE (Kingma & Welling, 2013), i.e., $\mathbf{z}_i = \text{Enc}(\mathbf{x}_i)$. In this setting, all latent frames share the same noise level $k$, and the reverse diffusion process restores the clean sequence by iteratively denoising:

$$p_\theta(\mathbf{z}_{0:T}^{k-1} \mid \mathbf{z}_{0:T}^k) = \mathcal{N}\big(\mathbf{z}_{0:T}^{k-1}; \mu_\theta(\mathbf{z}_{0:T}^k, k), \sigma_k^2 I\big), \tag{1}$$

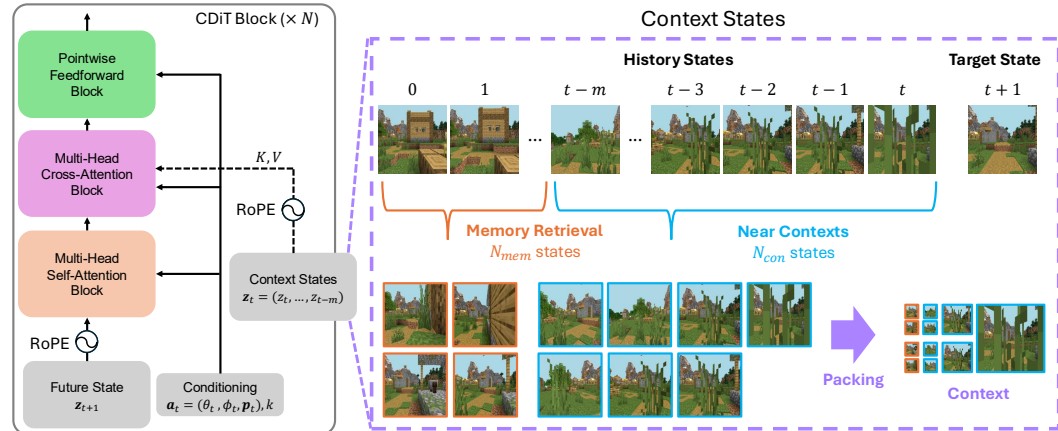

Figure 1: WorldPack consists of (1) CDiT with RoPE-based timestep embedding, (2) memory retrieval of the past states, and (3) packing the trajectory into the context.

where $\mathbf{z}_{0:T}^k$ denotes the noisy latent sequence at noise level $k$. This full-sequence formulation provides global guidance across frames, but constrains the sequence length to that used during training and lacks flexibility for long-horizon rollouts.

To overcome this limitation, we adopt an autoregressive formulation. Instead of generating the entire sequence jointly, the model conditions on the most recent $m$ latent frames to predict the next one:

$$p_\theta(\mathbf{z}_{t+1} \mid \mathbf{z}_{t-m+1:t}), \tag{2}$$

where generation proceeds sequentially. This setup naturally extends video length beyond the training horizon and supports long-term coherent generation.

Finally, to obtain an interactive video world model, we further introduce action sequences into the formulation. Given past latent states $\mathbf{z}_{t-m:t}$ and the current action $\mathbf{a}_t$, we learn a stochastic transition model $F_\theta$:

$$\mathbf{z}_{t+1} \sim F_\theta(\mathbf{z}_{t+1} \mid \mathbf{z}_{t-m:t}, \mathbf{a}_t). \tag{3}$$

This formulation approximates the environment dynamics $p(\mathbf{z}_{t+1} \mid \mathbf{z}_{\leq t}, \mathbf{a}_{\leq t})$, while operating in the compressed latent space. Predicted next state can then be decoded back to pixel space for visualization, enabling action-conditioned video generation and long-term world simulation.

## 4 WORLDPACK

WorldPack adopts a conditional diffusion transformer (CDiT) (Bar et al., 2024) as the backbone for history and action conditioning and incorporates RoPE-based temporal embeddings (Su et al., 2023), allowing effective use of memories regardless of temporal distance (Section 4.1). The compressed memory combines *memory retrieval* for consistent long-horizon reasoning (Section 4.2) from the past states and *trajectory packing* for context efficiency (Section 4.3).

### 4.1 VIDEO WORLD MODELING WITH CONDITIONAL DIFFUSION TRANSFORMER

Following Section 3, we design $F_\theta$ as a probabilistic mapping to simulate stochastic environments. To this end, we employ CDiT (Bar et al., 2024), which is a temporally autoregressive transformer model, and where efficient CDiT blocks are applied $N$ times over the input sequence (Figure 1). Unlike a standard Transformer that applies self-attention across all tokens, CDiT restricts self-attention to the tokens of the denoised target frame and incorporates cross-attention over past frames, allowing efficient learning. This cross-attention contextualizes the representation through skip connections, and conditioning on input actions is incorporated. While a standard DiT (Peebles & Xie, 2023) can be directly applied, its computational complexity scales quadratically with context length, i.e., $O(m^2 n^2 d)$ for $n$ tokens per frame, $m$ frames, and token dimension $d$. In contrast, CDiT is dominated by the cross-attention complexity $O(m n^2 d)$, which scales linearly with context length, enabling the use of longer contexts.

In addition, our model must integrate memory contexts located at arbitrary temporal distances from the current timestep. To achieve this, we adopt Rotary Position Embeddings (RoPE) (Su et al., 2023) as a position-aware design. RoPE enables consistent temporal representations regardless of variable context length, providing stable embeddings even for memory frames selected at arbitrary distances. This allows memory-aware inference over sequences with long-term dependencies.

## 4.2 MEMORY RETRIEVAL

Previous video world models that incorporated memory often design the importance of past frames based on the overlap of camera fields of view (Yu et al., 2025; Xiao et al., 2025). However, explicit camera fields of view are not always available, such as in real-world environments. Therefore, we generalize a scoring function so that we can predict frame importance solely from the position and orientation (yaw and pitch). We denote the current position as $\mathbf{p} = (x_t, y_t, 0)^\top$ and the viewing direction, computed from yaw $\theta_t$ and pitch $\phi_t$, as the unit vector:

$$\mathbf{d} = (\cos\phi_t \cos\theta_t,\ \cos\phi_t \sin\theta_t,\ \sin\phi_t)^\top. \tag{4}$$

For each past frame $i$, the agent's position is $\mathbf{p}_i = (x_i, y_i, 0)^\top$ and the corresponding direction is

$$\mathbf{d}_i = (\cos\phi_i \cos\theta_i,\ \cos\phi_i \sin\theta_i,\ \sin\phi_i)^\top. \tag{5}$$

Based on these, we compute:

$$s_i = (\mathbf{p}_i - \mathbf{p})^\top \mathbf{d}, \qquad \text{(forward projection)} \tag{6}$$
$$\ell_i = \|(\mathbf{p}_i - \mathbf{p}) - s_i\mathbf{d}\|, \qquad \text{(lateral distance)} \tag{7}$$
$$\cos\Delta\theta_i = \mathbf{d}_i^\top \mathbf{d}, \qquad \text{(directional similarity).} \tag{8}$$

The importance score for frame $i$ is then defined as

$$\begin{aligned}
\text{score}_i = {}& w_c \cdot \max(\cos\Delta\theta_i, 0)\ \exp\!\left(-\tfrac{s_i^2}{2\sigma_s^2}\right) \exp\!\left(-\tfrac{\ell_i^2}{2\sigma_\ell^2}\right) \\
& + w_a \cdot \max(-\cos\Delta\theta_i, 0)\ \exp\!\left(-\tfrac{(s_i-\mu_s)^2}{2\sigma_s^2}\right) \exp\!\left(-\tfrac{\ell_i^2}{2\sigma_\ell^2}\right).
\end{aligned} \tag{9}$$

In practice, to avoid redundancy, we introduce an exclusion window of 20 frames (equivalent to one second at 20 frames per second), ensuring that frames within this range are not selected solely based on their scores. This encourages the retrieved context to span a broader temporal range, preventing the model from overemphasizing temporally adjacent frames and allowing it to exploit long-term spatial information. We set the parameters to $\sigma_\ell = 10.0, \mu_s = 1.0, \sigma_s = 0.01, w_c = 1.0, w_a = 1.0$. This design prioritizes frames that are spatially close and aligned with the current view direction, while also incorporating opposite-facing frames at a characteristic distance. As a result, effective memory retrieval can be achieved even without explicit information about camera fields of view.

## 4.3 PACKING TRAJECTORY INTO CONTEXT

Previous video world models have been constrained by a fixed context length, which prevented them from incorporating long-term history. As a result, while they remained sensitive to recent observations, it was challenging to predict scenes that depend on events further in the past. This limitation caused errors to accumulate during rollouts, leading the generated trajectories to diverge from the original world gradually.

To overcome this issue, we propose trajectory packing. Trajectory packing enables efficient utilization of long-term history within a fixed-length context by hierarchically compressing and allocating trajectories. Specifically, past frames are encoded at different resolutions depending on their temporal distance: recent frames are preserved at high resolution, while older frames are compressed and stored at lower resolution. In addition, by incorporating memory retrieval, even frames beyond the nominal context length can be selectively integrated into the context if they are deemed essential. This design enables the model to simultaneously retain recent observations, long-term history, and salient memory elements, thereby allowing for reasoning over broad temporal scales during prediction.

Formally, let the recent past latent frames stored in memory be $\mathbf{z}_t, \mathbf{z}_{t-1}, \ldots, \mathbf{z}_{t-N_{\text{con}}}$, where $\mathbf{z}_t$ denotes the most recent frame and $\mathbf{z}_{t-N_{\text{con}}}$ the oldest. Here, $N_{\text{con}}$ represents the number of consecutive past frames maintained in the context window. In addition, we define memory frames as $\mathbf{z}_{M_1}, \mathbf{z}_{M_2}, \ldots, \mathbf{z}_{M_{N_{\text{mem}}}}$, which correspond to frames extracted from the history that are considered important, even beyond the nominal context length. Here, $N_{\text{mem}}$ denotes the number of retrieved memory frames. Trajectory packing handles both regular past latent frames and memory frames in a unified manner by applying hierarchical compression. Each past latent frame $\mathbf{z}_{t-i}$ and memory frame $\mathbf{z}_{M_j}$ is assigned an effective context length $\ell_{t-i}$ or $\ell_{M_j}$ after Transformer patchifying, with the compression rate determined by the temporal distance or importance of the frame:

$$\ell_{t-i} = \frac{L_f}{\lambda^i}, \quad \ell_{M_j} = \frac{L_f}{\lambda^{d_j}}, \tag{10}$$

where $L_f$ is the base context length for the most recent frame, $\lambda > 1$ controls how aggressively older or memory frames are compressed, and $d_j$ denotes the temporal distance or selection-based scale of the memory frame $\mathbf{z}_{M_j}$. For example, $\lambda = 2, i = 2$ corresponds to a $4 \times 4$ patchify kernel, while $i = 4$ corresponds to an $8 \times 8$ kernel. The total packed context length is then given by:

$$L_{\text{pack}} = S \cdot L_f + \sum_{i=S+1}^{N_{\text{con}}} \ell_{t-i} + \sum_{j=1}^{N_{\text{mem}}} \ell_{M_j}, \tag{11}$$

where $S$ denotes the number of uncompressed slots reserved for the most recent frames. This formulation ensures that recent frames are preserved at high resolution. In contrast, older and memory frames are progressively compressed, allowing the model to incorporate long-term history without incurring a linear increase in computational cost.

In practice, we represent frames more efficiently by applying geometric compression (Zhang & Agrawala, 2025). Specifically, we set compression ratios of $2^0$, $2^2$, and $2^4$, which correspond to context lengths of 1, 2, and 16, respectively, and train across a total of 19 context lengths. Additionally, we replace the last 8 frames with those selected by memory retrieval. This design allows recent frames to be preserved at high resolution. In contrast, older frames are compressed to lower resolution, enabling the model to retain long-term history while keeping computation efficient. Furthermore, to account for distributional differences across compression levels, we assign independent input projection layers for each compression ratio, rather than sharing a single projection. These layers are initialized by interpolating from the pretrained patchify layer of the base model with a kernel size of $(4, 4)$. As a result, the model achieves generalized temporal representations that can handle memory contexts selected from arbitrary historical contexts.

## 5 EVALUATION ON SPATIAL CONSISTENCY

We primarily focus on evaluating the ability of video world models to retain long-term spatial memory. For this purpose, we leverage Loop-Nav (Lian et al., 2025), a benchmark constructed in Minecraft environments. LoopNav is designed for loop-style navigation tasks, where the agent explores a portion of the environment and then returns to an earlier location within it. This design provides a precise and targeted method for testing whether a model can recall and reconstruct previously observed scenes, making LoopNav a distinctive benchmark for evaluating spatial memory.

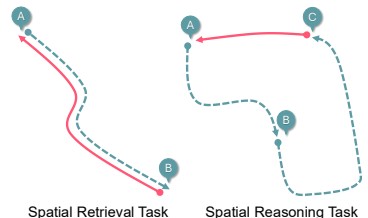

Figure 2: Illustration of the two LoopNav benchmark tasks. (**Left**) Spatial Memory Retrieval Task: the agent explores along A→B (blue path) and must reconstruct earlier observations on the return path B→A (red path). (**Right**) Spatial Reasoning Task: the agent explores along A→B→C (blue path) and must reconstruct the environment on the longer return path C→A (red path), requiring reasoning across accumulated spatial memory.

**Spatial Memory Retrieval Task (ABA).** The most basic setting of LoopNav is the A→B→A trajectory (Figure 2; **Left**). In this case, the segment from A to B acts as the exploration phase, supplying contextual observations to the model.

The return path from B to A constitutes the reconstruction phase, during which the model must demonstrate spatial consistency in regenerating observations from earlier locations. Because the ground-truth sequence has already been observed, this scenario is best viewed as a spatial retrieval task, explicitly probing whether the model can reproduce information embedded in the context.

**Spatial Reasoning Task (ABCA).** Here, A→B→C forms the exploration phase, while C→A is evaluated as the reconstruction phase (Figure 2; **Right**). Unlike an A→B→A loop, this task challenges the model to rely on accumulated spatial memory to reconstruct the environment along an extended path, potentially across areas observed from different viewpoints or at earlier time steps. This setup is closely related to a spatial reasoning task, where success requires leveraging contextual knowledge to generate coherent future observations rather than simply retrieving frames.

**Metrics.** For evaluation, we use LPIPS (Zhang et al., 2018) to assess semantic-level perceptual fidelity, and SSIM (Wang et al., 2004) to evaluate low-level structural alignment. We further employ DreamSim (Fu et al., 2023), which measures perceptual similarity based on deep feature representations, and PSNR to capture pixel-level reconstruction quality. Since no single metric fully reflects semantic accuracy or long-term spatial coherence, we complement these quantitative results with qualitative inspection by human observers.

# 6 EXPERIMENTS

## 6.1 EXPERIMENTAL SETUP

**Baselines.** Oasis (Decart et al., 2024) is a world model that employs a ViT (Dosovitskiy et al., 2020) as a spatial autoencoder and a DiT (Peebles & Xie, 2023) as the latent diffusion backbone, trained with Diffusion Forcing (Chen et al., 2024). It generates frames autoregressively with user-controllable conditioning, and the publicly available Oasis-500M model is evaluated with a context length of 32. Mineworld (Guo et al., 2025) is an interactive world model based on a pure Transformer architecture, generating new scenes from paired game frames and actions, with its pretrained checkpoint evaluated at a context length of 15. DIAMOND (Alonso et al., 2024) is a diffusion-based world model built upon a UNet architecture (Ronneberger et al., 2015), generating frames conditioned on past observations and actions, and evaluated with a context length of 4. NWM (Bar et al., 2024) is a controllable video generation model that predicts future observations conditioned on navigation actions, leveraging CDiT with a context length of 4.

## 6.2 RESULTS

In the multi-step rollout generation (Table 1 and Table 2), WorldPack, despite shortest context length, outperforms the baselines – Oasis, Mineworld, DIAMOND, and NWM – in SSIM and LPIPS, and also surpasses NWM in PSNR and DreamSim, and FVD. However, the results for SSIM were not decisively superior, remaining only partially competitive. This tendency can be explained by the inherent limitations of distortion-based metrics, which favor spatially averaged or blurred predictions that minimize pixel-wise differences while sacrificing perceptual fidelity (Blau & Michaeli, 2018). Indeed, Lian et al. (2025) also reported that SSIM exhibits only a weak correlation with perceptual quality in visualizations. In addition, qualitative evaluations confirmed that WorldPack maintains long-term consistency, showing only minor deviations from the ground truth even when rollouts are extended (Figure 3).

Taken together, these results demonstrate consistent improvements across both the ABA and ABCA tasks, in terms of both quantitative metrics and qualitative assessments. In particular, the proposed compressed memory mechanism plays a crucial role in achieving high context efficiency and maintaining long-term spatial consistency, even under shortest context lengths.

## 6.3 ABLATION STUDY

To examine the effect of memory retrieval, we focus on cases where prediction becomes difficult using only the most recent frames. Without memory retrieval, trajectory packing compresses only the most recent $N_{\text{con}}$ frames from the past context and uses them as input. However, in both the ABA and ABCA tasks, this setting loses critical cues needed to predict the terminal BA and CA segments,

Table 1: Model performance on tasks of varying type and difficulty. ABA denotes the spatial memory retrieval tasks, and ABCA denotes the spatial reasoning tasks. The navigation range (5, 15, 30, 50) indicates the size of the area within which the agent is required to move. SSIM (↑) evaluates better structural consistency, while LPIPS (↓) reflects perceptual fidelity. We refer to baseline evaluation results from Lian et al. (2025).

| Nav. Range | Model | Context | Trajectory | SSIM ↑ | | LPIPS ↓ | |
|---|---|---|---|---|---|---|---|
| | | | | ABA | ABCA | ABA | ABCA |
| 5 | Oasis | 32 | 32 | 0.36 | 0.34 | 0.76 | 0.82 |
| | Mineworld | 15 | 15 | 0.31 | 0.32 | 0.73 | 0.72 |
| | DIAMOND | 4 | 4 | **0.40** | **0.37** | 0.75 | 0.79 |
| | NWM | 4 | 4 | 0.33 | 0.31 | 0.64 | 0.67 |
| | WorldPack (ours) | 2.84 | 19 | 0.39 | 0.35 | **0.52** | **0.56** |
| 15 | Oasis | 32 | 32 | 0.37 | 0.38 | 0.82 | 0.81 |
| | Mineworld | 15 | 15 | 0.34 | 0.32 | 0.74 | 0.74 |
| | DIAMOND | 4 | 4 | 0.38 | 0.39 | 0.78 | 0.79 |
| | NWM | 4 | 4 | 0.30 | 0.33 | 0.67 | 0.65 |
| | WorldPack (ours) | 2.84 | 19 | **0.48** | **0.46** | 0.57 | 0.55 |
| 30 | Oasis | 32 | 32 | 0.33 | 0.35 | 0.86 | 0.85 |
| | Mineworld | 15 | 15 | 0.33 | 0.28 | 0.77 | 0.77 |
| | DIAMOND | 4 | 4 | **0.37** | **0.35** | 0.81 | 0.81 |
| | NWM | 4 | 4 | 0.32 | 0.30 | 0.69 | 0.71 |
| | WorldPack (ours) | 2.84 | 19 | 0.32 | 0.28 | **0.61** | **0.63** |
| 50 | Oasis | 32 | 32 | 0.36 | 0.36 | 0.86 | 0.83 |
| | Mineworld | 15 | 15 | 0.31 | 0.32 | 0.78 | 0.75 |
| | DIAMOND | 4 | 4 | **0.37** | **0.38** | 0.83 | 0.81 |
| | NWM | 4 | 4 | 0.28 | 0.33 | 0.72 | 0.65 |
| | WorldPack (ours) | 2.84 | 19 | 0.27 | 0.31 | **0.63** | **0.63** |

Table 2: Evaluation of models on spatial memory (ABA) and reasoning (ABCA) tasks under different navigation ranges. PSNR (↑) reflects pixel-level reconstruction accuracy, DreamSim (↓) captures perceptual similarity based on deep features, and FVD (↓) measures temporal video quality.

| Nav. Range | Model | Context | Trajectory | PSNR ↑ | | DreamSim ↓ | | FVD ↓ | |
|---|---|---|---|---|---|---|---|---|---|
| | | | | ABA | ABCA | ABA | ABCA | ABA | ABCA |
| 5 | NWM | 4 | 4 | 12.3 | 10.0 | 0.33 | 0.44 | **747** | 759 |
| | WorldPack (ours) | 2.84 | 19 | **12.6** | **11.1** | **0.30** | **0.35** | 760 | **670** |
| 15 | NWM | 4 | 4 | 11.5 | 11.5 | 0.44 | 0.38 | 665 | 773 |
| | WorldPack (ours) | 2.84 | 19 | **12.0** | **11.7** | **0.40** | **0.36** | **551** | **669** |
| 30 | NWM | 4 | 4 | 11.1 | 10.0 | 0.45 | 0.49 | 755 | 819 |
| | WorldPack (ours) | 2.84 | 19 | **11.3** | **11.1** | **0.41** | **0.42** | **570** | **679** |
| 50 | NWM | 4 | 4 | 10.2 | 9.8 | 0.47 | 0.48 | 841 | 810 |
| | WorldPack (ours) | 2.84 | 19 | **10.7** | **10.5** | **0.42** | **0.41** | **562** | **455** |

and the performance degradation becomes particularly severe when the navigation range is large. To evaluate this effect, we measured prediction accuracy on the terminal frames of trajectories in the LoopNav benchmark. Figure 4; **Top** shows the prediction performance on the last 61 frames in the ABCA task with navigation range = 30, while Figure 4; **Bottom** shows the performance on the last 101 frames with navigation range = 50. In both cases, we compare three settings: base model (no compressed memory), trajectory packing only, and trajectory packing combined with memory retrieval.

The results show that trajectory packing alone brings only marginal improvements over base model, merely benefiting from extended access to recent frames. In contrast, incorporating memory retrieval leads to a substantial performance gain. This indicates that by enriching the compressed context with retrieval-based information, the model can selectively exploit scene cues that are not contained in the most recent frames but are essential for accurate prediction. These results clearly demonstrate

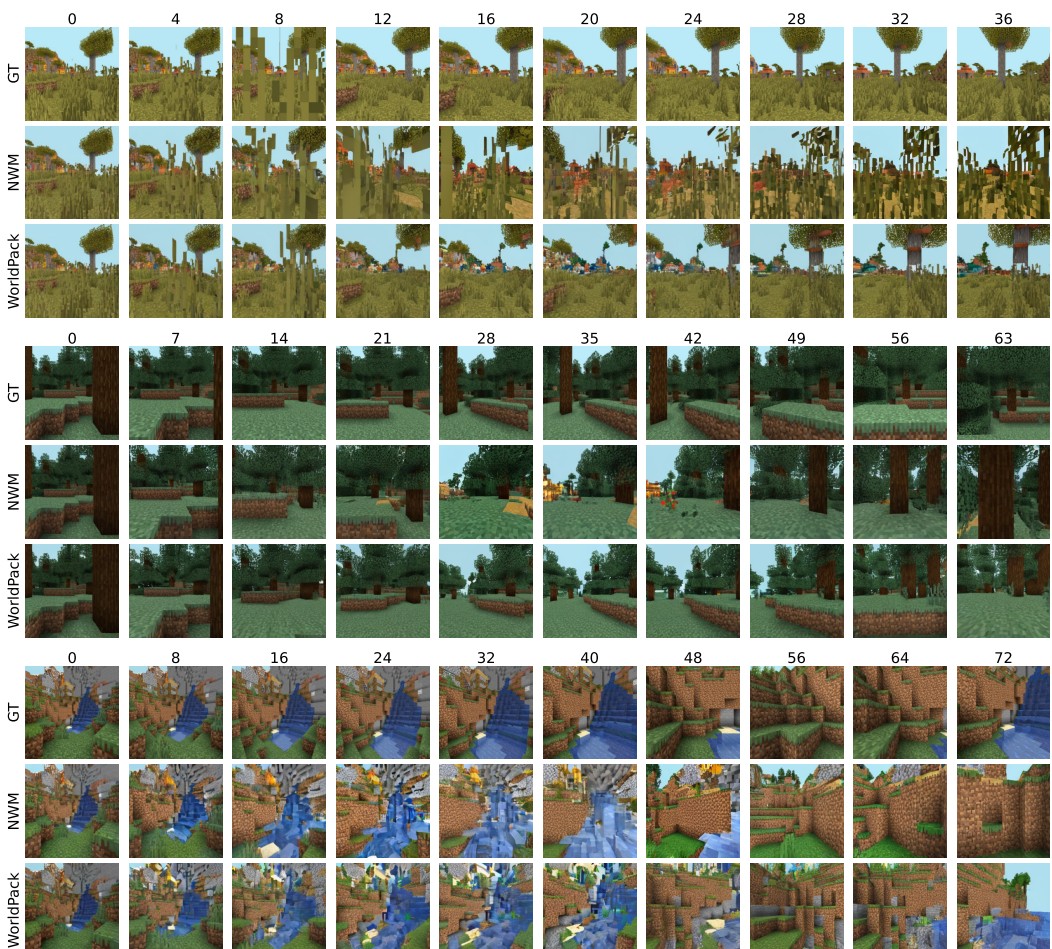

Figure 3: Visualization of rollouts. We compare ground truth (GT), NWM (Bar et al., 2024), and WorldPack. WorldPack can predict more similar states than NWM, especially in the latter part of the rollouts.

that memory retrieval is an indispensable component for achieving long-term spatial consistency and high-quality predictions. Results for the terminal frame prediction performance on other trajectories are provided in the Appendix A.

Next, we compare the performance when adopting only one of the two components of WorldPack, namely, trajectory packing or memory retrieval. The comparison is conducted under the ABA task with navigation range = 5. When using trajectory packing only, the most recent 19 trajectories are compressed into a context of size 2.84. In contrast, when using memory retrieval only, the model utilizes the most recent 1 trajectory together with 3 retrieved memories, resulting in a context of size 4 without packing. As shown in Figure 5, both packing-only and memory-only settings yield improvements over the base model, but the gains remain limited. In contrast, combining the two components achieves the most substantial performance improvements. This result indicates that both efficient long-term context retention via trajectory packing and the selective retrieval of important frames beyond the recent context are indispensable for world modeling that requires long-term spatial memory awareness.

## 6.4 Experiments with Real-World Data

To verify the practical usefulness of WorldPack beyond simulator environments such as Minecraft, we conducted experiments using real-world data. Specifically, we evaluated our method on the RECON dataset (Shah et al., 2021), one of the commonly used datasets in prior video-generation world model studies (Shah et al., 2022; Sridhar et al., 2024; Bar et al., 2024).

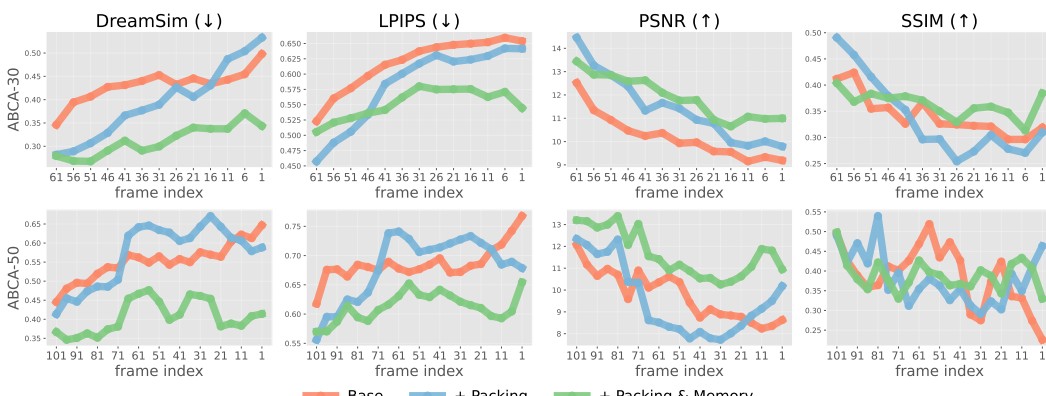

Figure 4: Prediction performance on the terminal frames of ABCA trajectories with different navigation ranges. **Top**: last 61 frames in ABCA-30. **Bottom**: last 101 frames in ABCA-50. We compare base model (no compressed memory), trajectory packing only, and trajectory packing + memory retrieval. Incorporating memory retrieval leads to substantial improvements, demonstrating that the model can exploit informative cues beyond the most recent frames.

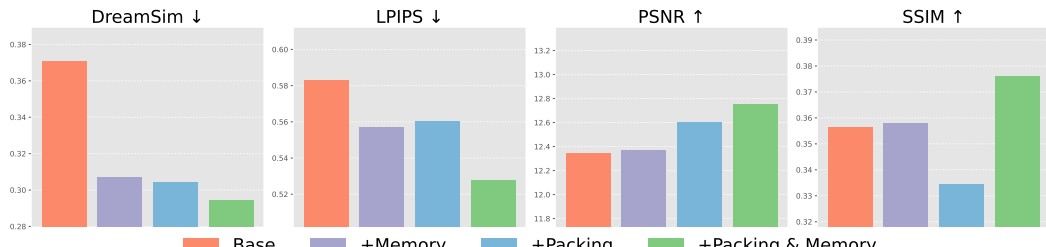

Figure 5: Comparison of using trajectory packing only, memory retrieval only, and their combination in WorldPack (ABA task, navigation range = 5). In the trajectory packing–only setting, the most recent 19 trajectories are compressed into a context of size 2.84. In the memory retrieval–only setting, the most recent 1 trajectory and 3 retrieved memories are used, yielding a context of size 4 without packing. While either component alone provides modest improvements over the base model, the largest performance gain is obtained when both are combined, demonstrating that the two mechanisms are essential for world modeling with long-term spatial memory awareness.

In our experiments, we used the first 20 frames as context and generated the subsequent frames. The quantitative results are shown in Table 3. These results demonstrate that WorldPack achieves strong generative performance even on real-world data, confirming its effectiveness beyond simulated environments.

### 6.5 ANALYSIS OF COMPUTATIONAL EFFICIENCY

We present the single-step inference time and memory costs for the diffusion model (Table 4). Compared to the baseline, WorldPack significantly extends the visible length of past trajectories from 4 to 19 frames. Although the incorporation of memory compression and retrieval processes introduces a slight overhead, the increase in inference time is marginal, at approximately 9%. Notably, memory consumption is actually reduced; this is because the compression mechanism lowers the number of tokens input into the CDiT (reducing the effective context from 4 frames to 2.84 frames, as shown in the context column). These experimental results corroborate WorldPack's ability to handle longer trajectory lengths with high computational efficiency.

## 7 DISCUSSION AND LIMITATION

Our evaluation is conducted within simulator environments, under the assumption that they are sufficient to assess the spatial memorization capability of video world models. We demonstrate both qualitative and quantitative improvements across spatial memorization tasks and spatial reasoning

Table 3: Evaluation of models on RECON dataset, real-world generation performance. Metrics include DreamSim (↓), LPIPS (↓), PSNR (↑), and SSIM (↑).

| Model | Context | Trajectory | DreamSim ↓ | LPIPS ↓ | PSNR ↑ | SSIM ↑ |
|---|---|---|---|---|---|---|
| NWM | 4 | 4 | 0.23 | 0.48 | 12.7 | 0.36 |
| + Packing | 2.84 | 19 | 0.18 | 0.45 | 13.4 | 0.40 |
| + Packing & Memory | 2.84 | 19 | **0.17** | **0.44** | **13.6** | **0.40** |

Table 4: Inference time and memory usage comparison.

| Model | Context | Trajectory | Inference Time (1-step, sec) | Memory Usage (GB) |
|---|---|---|---|---|
| Baseline | 4 | 4 | 0.430 | 22.08 |
| WorldPack | 2.84 | 19 | 0.468 | 21.78 |

tasks. Looking forward, it is essential to extend beyond simulator environments and incorporate real-world data (Yang et al., 2023; Wu et al., 2022). In this study, we primarily focused on the simulation ability of video world models and, therefore, evaluated their scene generation performance. As a future direction, we believe that exploring policy learning and planning with video world models (Alonso et al., 2024) will further deepen the discussion on the utility of spatial memory capabilities.

## 8 CONCLUSION

In this paper, we introduce WorldPack, a long-context-aware video world model through context compression. Memory retrieval module facilitates scene generation by selectively utilizing non-recent contextual spatial information. Trajectory packing enables the retention of long-term information without increasing computational costs by compressing past observations. We hope that this study will further promote the handling of long-context memory in video world models.

THE USE OF LARGE LANGUAGE MODELS

In this paper, we used LLMs mainly to polish writing and to propose paraphrasing.

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

# APPENDIX

## A  PREDICTION PERFORMANCE FOR LAST FRAMES

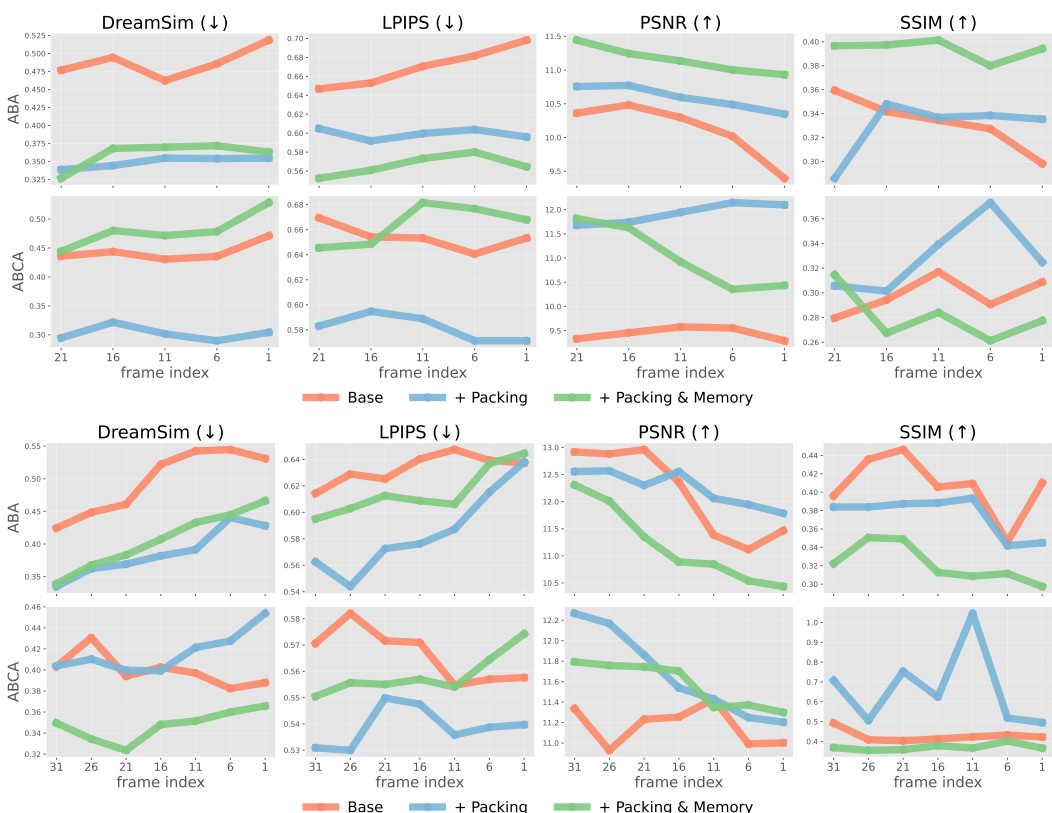

Figure 6: Prediction performance on the terminal frames of ABCA trajectories with different navigation ranges. **Top**: last 21 frames in ABA-5 and ABCA-5. **Bottom**: last 31 frames in ABA-15 and ABCA-15. We compare base model (no compressed memory), trajectory packing only, and trajectory packing + memory retrieval. Incorporating memory retrieval leads to substantial improvements, demonstrating that the model can exploit informative cues beyond the most recent frames.

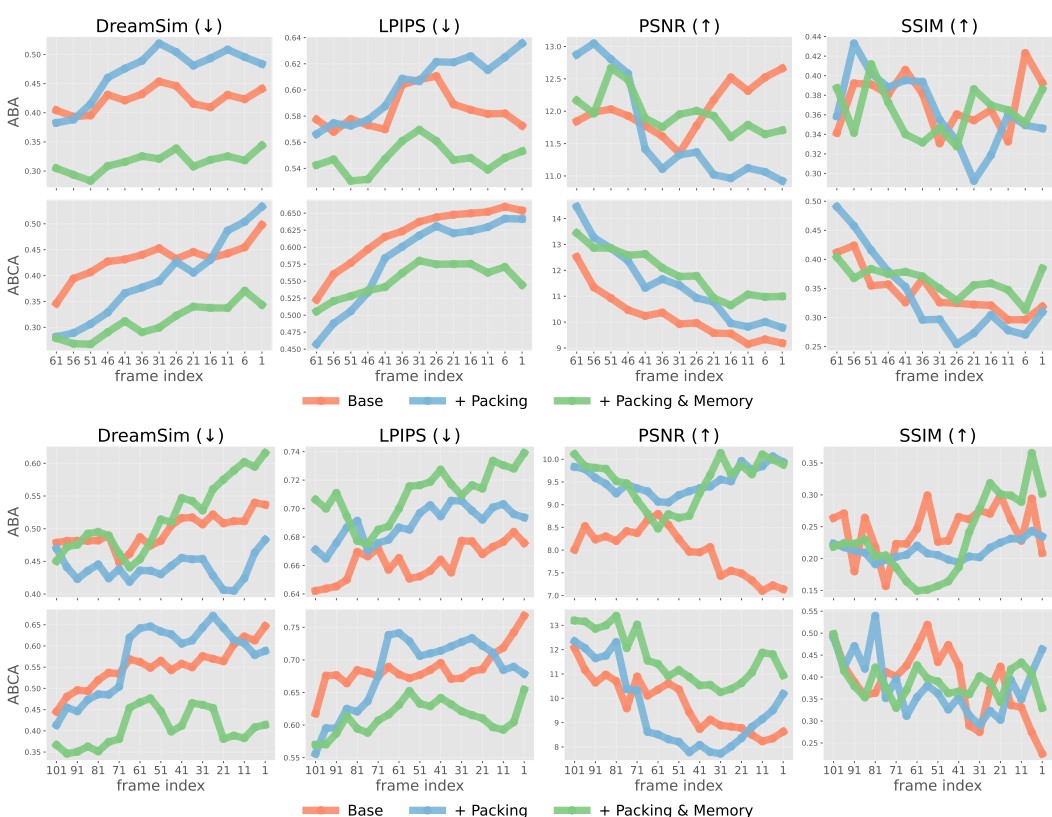

Figure 7: Prediction performance on the terminal frames of ABCA trajectories with different navigation ranges. **Top**: last 61 frames in ABA-30 and ABCA-30. **Bottom**: last 101 frames in ABA-50 and ABCA-50. We compare base model (no compressed memory), trajectory packing only, and trajectory packing + memory retrieval. Incorporating memory retrieval leads to substantial improvements, demonstrating that the model can exploit informative cues beyond the most recent frames.

