# OpenReview forum: "WorldPack: Compressed Memory Improves Spatial Consistency in Video World Modeling"
_ICLR.cc/2026/Conference — Submitted to ICLR 2026_

### Official Review · Reviewer_wZPH · 2025-10-21

**Soundness:** 2
**Presentation:** 3
**Contribution:** 2
**Rating:** 4
**Confidence:** 4

**Summary:**

This paper introduces WorldPack, a video world model targetting on long-term spatial consistency. According to the authors, there are two key components: trajectory packing, which compresses past frames hierarchically to retain long-term information efficiently; memory retrieval, which selectively recalls relevant past scenes for better spatial understanding. The evaluatiion on LoopNav (a benchmark on Minecraft) shows that WorldPack clearly outperforms prior methods including Oasis, MineWorld, DIAMOND, and NWM.

**Strengths:**

1. Leveraging the idea of FramePack is effective in enabling long video generation, which significantly reduces the context and improves the context efficiency. The constant memory is a good attribute for long, auto-regressive video generation.
2. The experiment on LoopNav is impressive.
3. The drawing is good and clear.

**Weaknesses:**

1. Firstly, I'm not clear about the performance gained from the FramePack design, which aims for better / longer video generation. But in your experiments, the number of frames in the generated videos are similar with NWM, can you show some comparisons with NWM on longder video generations? Like minutes with early 1,000 frames.
2. The FramePack design has a clear drawback is that the importance is manually assigned rather than learned from data. Can you add some additional discussions on that? Why in the field of NWM, this manually assigned importance works well or at least better than prior methods?
3. Secondly, though the results on LoopNav is promising, the experiments on real-world scenarios is missing, e.g., Go Stanford. I'm absolutely preferring a real-world benchmark than a Minecraft benchmark, which is much more practical.
4. Following 1, I encourage the authors to conduct the experiments between WorldPack and NWM on long, real-world cases, and report the results following the Figure 4 of NWM, and of course, the x-axis should represent longer period. I think it will be more clear to demonstrate the improved "long-term spatial consistency" over NWM.
5. The generated videos on both real-world and Minecraft cases is missing. Can you provide some videos compressed in *.zip?
6. The FVD in LoopNav is missing.
7. For the writing, I don't really get why "Compressed Memory Improves Spatial Consistency". This is anti-intuitive. You can claim your method brings compressed memory and improved spatial consistency if your experiments well support that.
8. Is the efficiency on running time also compared with NWM? I saw only the "context efficiency".

**Questions:**

Please see the Weaknesses*.

---

> ### Author Response · Authors · 2025-11-21
>
> We thank you for constructive feedback.
>
> > W2
>
> We agree with the reviewer that relying solely on manually assigned importance is insufficient for robust world modeling. To address this, we employed the Memory Retrieval mechanism proposed in Section 4.2.
>
> By combining Memory Retrieval with Trajectory Packing, our method utilizes the agent's positional history to dynamically calculate which frames require attention. This allows the model to selectively determine the degree of compression for each frame in the history. We confirmed in Figure 5 that this mechanism achieves higher world modeling performance compared to designs that simply allocate manual importance to recent frames.
>
> Furthermore, to validate our method on more realistic domains, we conducted additional evaluations on the RECON dataset [1], a standard benchmark in prior video-generation world model studies [2, 3, 4]. As shown in the table below, the + Packing & Memory configuration outperforms the baselines, demonstrating the efficacy of our approach (please compare "+ Packing" and "+ Packing & Memory").
>
> | Model                   | Context | Trajectory | DreamSim ↓ | LPIPS ↓ | PSNR ↑ | SSIM ↑ |
> |-------------------------|---------|------------|-------------|---------|--------|---------|
> | NWM                     | 4       | 4          | 0.23        | 0.48    | 12.7   | 0.36    |
> | + Packing               | 2.84    | 19         | 0.18        | 0.45    | 13.4   | 0.40    |
> | + Packing & Memory      | 2.84    | 19         | **0.17**    | **0.44**| **13.6**| **0.40** |
>
> [1] Shah, et al. Rapid Exploration for Open-World Navigation with Latent Goal Models. CoLR2021.
> [2] Shah, et al. GNM: A General Navigation Model to Drive Any Robot. ICRA2023.
> [3] Sridhar, et al. Nomad: Goal masked diffusion policies for navigation and exploration. ICRA2024.
> [4] Bar, et al. Navigation World Models. CVPR2025.
>
>
> > W1、W3、W4
>
> To verify the practical usefulness of WorldPack beyond simulator environments such as Minecraft, we conducted experiments using real-world data.
> Specifically, we evaluated our method on the RECON dataset [1], one of the commonly used datasets in prior video-generation world model studies [2, 3, 4] (please see W2).
> In our experiments, we used the first 20 frames as context and generated the subsequent frames.
> The quantitative results are shown in \autoref{tab:multi_real}.
> These results demonstrate that WorldPack achieves strong generative performance even on real-world data, confirming its effectiveness beyond simulated environments.
>
> Due to computational resource constraints, experiments involving longer-term verification and wider trajectory inputs are currently in progress. We expect to report these additional results within the next week.
>
> [1] Shah, et al. Rapid Exploration for Open-World Navigation with Latent Goal Models. CoLR2021.
> [2] Shah, et al. GNM: A General Navigation Model to Drive Any Robot. ICRA2023.
> [3] Sridhar, et al. Nomad: Goal masked diffusion policies for navigation and exploration. ICRA2024.
> [4] Bar, et al. Navigation World Models. CVPR2025.
>
> > W5
>
> We have provided the generated videos in the supplementary zip file.
>
> > W6
>
> We have added Fréchet Video Distance (FVD) results to provide stronger evidence of our model's performance. As shown in the table below, WorldPack outperformed the baseline NWM in the majority of settings, with the exception of short-term horizons. These experimental results reinforce the evidence that WorldPack is highly effective for video world modeling tasks.
>
> | Nav. Range | Model | FVD (ABA) | FVD (ABCA) |
> |--|--|--|--|
> | 5 | NWM | **747** | 759 |
> | | WorldPack (ours) | 760 | **670** |
> | 15 | NWM | 665 | 773 |
> | | WorldPack (ours) | **551** | **669** |
> | 30 | NWM | 755 | 819 |
> | | WorldPack (ours) | **570** | **679** |
> | 50 | NWM | 841 | 810 |
> | | WorldPack (ours) | **562** | **455** |
>
> > W7
>
> The primary reason Packing enhances temporal consistency is its ability to condition on a significantly longer temporal context.
>
> Both the baseline NWM and the "Memory-only" configuration are constrained to a token budget equivalent to 4 frames, effectively limiting their visibility to only 4 past frames (please refer to the "Context" and "Trajectory" columns in the table in W8). Specifically, the baseline utilizes the 4 most recent frames, while the Memory model utilizes 1 recent frame combined with 3 retrieved frames.
>
> In contrast, the introduction of Packing allows us to encode information from 19 frames into a context equivalent to only 2.84 frames. In the Packing + Memory setting, this context comprises 11 recent frames and 8 retrieved frames. Therefore, despite the reduced token count, the model gains access to information from a much larger number of frames. We attribute the improvement in temporal consistency to this increased information density.

---

> ### Author Response · Authors · 2025-11-21
>
> > W8
>
> We present the single-step inference time and memory costs for the diffusion model below.
>
> Compared to the baseline, WorldPack significantly extends the visible length of past trajectories from 4 to 19 frames. Although the incorporation of memory compression and retrieval processes introduces a slight overhead, the increase in inference time is marginal, at approximately 9%. Notably, memory consumption is actually reduced; this is because the compression mechanism lowers the number of tokens input into the CDiT (reducing the effective context from 4 frames to 2.84 frames).
>
> These experimental results corroborate WorldPack's ability to handle longer trajectory lengths with high computational efficiency.
>
> | Model | Context | Trajectory | Inference Time (1-step, sec) | Memory Usage (GB) |
> |--|--|--|--|--|
> | Baseline | 4 | 4 | 0.430 | 22.08 |
> | WorldPack | 2.84 | 19 | 0.468 | 21.78 |

---

### Official Review · Reviewer_42nA · 2025-10-28

**Soundness:** 2
**Presentation:** 2
**Contribution:** 2
**Rating:** 2
**Confidence:** 4

**Summary:**

This paper focuses on the challenge of long-horizon generation in video world models.

This paper introduces WorldPack, which achieves long-term temporal and spatial consistency through an efficient compressed memory design.

On the LoopNav benchmark in Minecraft, WorldPack demonstrates good performance.

**Strengths:**

Introduces a compressed memory approach (trajectory packing + memory retrieval) that creatively addresses long-horizon consistency without increasing context length.

Empirically outperforms baselines on LoopNav, improving spatial consistency, fidelity, and long-term generation quality.

**Weaknesses:**

**Limited innovation in memory design**: The proposed trajectory packing and memory retrieval are common ideas that have been widely used, including in RNNs and LSTMs via memory banks. The paper does not articulate a fundamentally new algorithmic principle or provide a theoretically grounded formulation that distinguishes its approach from prior methods. Moreover, the related work section lacks a thorough discussion of these connections.

**Incremental engineering on existing backbones**: Using a CDiT backbone with RoPE-based temporal embeddings is a straightforward integration of known components that enhance long-range conditioning in Transformers. The gains appear to stem from combining established techniques rather than introducing a new architecture or training objective that advances the state of the art conceptually. These do not constitute genuine novelty for the paper.

**Missing references**: In the Video World Models section of related work, there has already been substantial exploration of long-term context in autonomous driving, yet the authors overlook these approaches [1,2]. Why is that? Was it an oversight, or a deliberate omission?

**Results**: The results in Table 1 are not particularly impressive. Moreover, the last two panels of ABCA-50 in Figure 4 suggest that the memory component may have limited impact; in fact, adding memory appears to underperform compared to not using it, especially over long horizons.

[1] DrivingWorld: Constructing World Model for Autonomous Driving via Video GPT

[2] InfinityDrive: Breaking Time Limits in Driving World Models

**Questions:**

1. Do the authors have better results to demonstrate the superiority of their method?

2. Could the author explain their contribution and novelty more clearly, as well as their previous work, and the difference between their network structure and theirs (using the memory bank in RNN and LSTM is not a novelty in my opinion).

---

> ### Author Response · Authors · 2025-11-21
>
> Thank you for your insightful reviews.
>
> > W1、W2、Q2
>
> Our primary technical contribution lies in investigating methods to efficiently incorporate past historical information into the context of world models based on Video Diffusion Models, not SSM-based world models.Because packing is fundamentally a mechanism for manipulating the Transformer’s context structure, it has not been explored within SSM-based world models, whose architectures do not rely on explicit context windows.
>
> Specifically, our contributions are threefold: the integration of CDiT with RoPE (Section 4.1), Memory Retrieval (Section 4.2), and Trajectory Packing (Section 4.3). The existing FramePack design suffers from a clear drawback: frame importance is manually assigned rather than being learned from data. To address this, we combine Memory Retrieval and Trajectory Packing to calculate which frames require attention based on the agent's positional history. This allows the model to selectively determine the degree of compression for each frame in the history.
>
> Furthermore, while prior work has not carefully analyzed architectural factors such as CDiT’s RoPE design, our study reveals that adjusting the RoPE formulation is indispensable for long-horizon world modeling. In contrast to earlier methods, we demonstrate that the default RoPE configuration fundamentally limits temporal range.
>
> Figure 5 confirms that this mechanism achieves higher world modeling performance compared to designs that simply allocate manual importance to recent frames. We further validated this effectiveness on a more realistic domain dataset [1] (please refer to the comparison between the "+ Packing" and "+ Packing & Memory" settings).
>
> | Model | Context | Trajectory | DreamSim ↓ | LPIPS ↓ | PSNR ↑ | SSIM ↑ |
> |--|--|--|--|--|--|--|
> | NWM | 4 | 4 | 0.23 | 0.48 | 12.7 | 0.36 |
> | + Packing | 2.84 | 19 | 0.18 | 0.45 | 13.4 | 0.40 |
> | + Packing & Memory | 2.84 | 19 | **0.17** | **0.44** | **13.6** | **0.40** |
>
> Finally, it is worth emphasizing that directly inserting all historical frames into the context window is computationally infeasible, given the quadratic complexity of attention and the rapidly increasing token count in long-horizon video generation. Our approach directly addresses this limitation by introducing mechanisms that enable world models to process substantially larger historical token sequences than existing methods, without sacrificing efficiency. We believe that this capability, allowing diffusion-based world models to scale to long-term histories through a principled combination of Retrieval, Packing, and RoPE modification, constitutes a meaningful and novel technical contribution to the design of long-horizon video world models.
>
> [1] Shah, et al. Rapid Exploration for Open-World Navigation with Latent Goal Models. CoLR2021.
>
>
> > W3
>
> Thank you for pointing out this reference. We have cited them in Section 2.
>
> > W4、Q1
>
> We added the results of FVD to show stronger evidence.
> We confirmed that WorldPack outperformed the NWM baseline in the majority of settings, with the exception of short-term configurations.
> These results reinforce the evidence that WorldPack is highly effective for long-term video world modeling.
>
> | Nav. Range | Model | FVD (ABA) | FVD (ABCA) |
> |--|--|--|--|
> | 5 | NWM | **747** | 759 |
> | | WorldPack (ours) | 760 | **670** |
> | 15 | NWM | 665 | 773 |
> | | WorldPack (ours) | **551** | **669** |
> | 30 | NWM | 755 | 819 |
> | | WorldPack (ours) | **570** | **679** |
> | 50 | NWM | 841 | 810 |
> | | WorldPack (ours) | **562** | **455** |

---

> > ### Comment · Reviewer_42nA · 2025-11-27
> >
> > Thank you for the authors' detailed replies. While they resolved most of my concerns, I still have some questions. The packing operation in Figure 1 doesn't seem to be a very good strategy. It's obvious that a lot of information is lost. Therefore, the efficiency advantage brought by packing might not be significant, and it might even result in a performance loss, especially when the frames retrieved from memory are crucial and highly overlap with the generated viewpoint. Therefore, I think [1] is a good approach, and your packing approach is somewhat redundant.
> >
> > Secondly, the statement "However, explicit camera fields of view are not always available, such as in real-world environments" is the essence of the authors' 4.2 contribution. However, in reality, whether in embodied intelligence (robots have many cameras), autonomous driving (Tesla's FSD), or Minecraft, obtaining camera parameters is a simple matter. Could you explain in which situations we cannot obtain camera parameters and list relevant references?
> >
> > [1] WORLDMEM: Long-term Consistent World Simulation with Memory
> > Zeqi Xiao, Yushi Lan, Yifan Zhou, Wenqi Ouyang, Shuai Yang, Yanhong Zeng, Xingang Pan

---

> > > ### Author Response · Authors · 2025-11-30
> > >
> > > Thank you for your additional comment.
> > >
> > > > Q1. Concerns regarding the packing operation (information loss vs. efficiency) and comparison with WorldMem [1].
> > >
> > > We thank the reviewer for the valuable feedback and for pointing out the reference [1]. We agree with the reviewer and [1] that retrieving and utilizing past frames from memory is crucial for maintaining long-term consistency.
> > >
> > > However, our proposed "packing" operation is not intended to conflict with memory retrieval; rather, it is a complementary strategy designed to maximize the effective context length within limited computational resources.
> > >
> > > As the reviewer noted, packing involves a trade-off that reduces spatial resolution. However, we hypothesize that, for long-term consistency, attending to a longer temporal history (even at reduced resolution) is more critical than maintaining full resolution over a shorter history.
> > >
> > > Indeed, our experimental results (Figure 5) demonstrate that the "Memory + Packing" model achieves higher performance than "Memory Retrieval only (without Packing)." This suggests that packing expands the receptive field, enabling the model to leverage past information more effectively. Therefore, we believe that applying packing in addition to approaches such as [1] is not redundant but rather highly effective for improving efficiency.
> > >
> > > > Q2. Clarification on the availability of explicit camera fields of view.
> > >
> > > We appreciate the comment. We agree that obtaining accurate camera parameters is straightforward in specific controlled environments such as robotics (with calibrated sensors), autonomous driving (e.g., FSD), and simulators (e.g., Minecraft).
> > >
> > > However, the essence of our contribution in Section 4.2 lies in building a general-purpose world model based on "minimal assumptions," capable of operating even when such geometric priors are unavailable or unreliable. We believe that eliminating the dependency on camera parameters offers the following advantages:
> > >
> > > Utilization of Internet-scale Video Data: When pre-training models using the vast amount of "in-the-wild" videos available on platforms like YouTube, most of the data lack camera-parameter annotations. By assuming "no parameters," we can directly utilize these rich datasets for training without the cost of estimation.
> > >
> > > Robustness to Uncalibrated Environments: In scenarios involving consumer devices without strict calibration (e.g., smartphones, AR glasses) or environments where SLAM might fail, avoiding a strong reliance on explicit parameters is desirable for robust control.
> > >
> > > Therefore, our argument is not merely about cases where parameters cannot be obtained, but rather that a model capable of learning and controlling without parameters offers greater data scalability and real-world versatility.

---

### Official Review · Reviewer_W2JJ · 2025-10-31

**Soundness:** 3
**Presentation:** 3
**Contribution:** 2
**Rating:** 4
**Confidence:** 4

**Summary:**

This paper proposes WorldPack, a video world model with compressed memory. Specifically, the method retrieves relevant past frames, then downsamples them to lower resolutions according to their estimated importance — a technique inspired by FramePack used in video generation. Experiments are conducted on LoopNav, a Minecraft-based benchmark designed to evaluate long-term cycle consistency.

**Strengths:**

1. The proposed method is technically sound, and the experimental results clearly demonstrate its superiority over provided baseline approaches.
2. The paper is well written and easy to follow.

**Weaknesses:**

The technical novelty of the paper is somewhat limited. The central idea, importance-based frame compression, has already been explored in next-frame prediction and video diffusion models (FramePack). The paper’s main contribution appears to be in adapting this idea to the world modeling context through specific retrieval strategies and benchmark evaluations.

- I suggest that FramePack be clearly highlighted in the Preliminaries section to better situate this work within existing literature and make its incremental contributions more transparent.

**Questions:**

1. Could the authors clarify the key differences between FramePack and trajectory packing?
2. In Section 4.2, it is stated that the proposed method does not require explicit camera parameters. However, aren’t Euler angles of the camera still considered part of its parameters? Please elaborate.
3. Why does WorldPack not include comparisons with memory-related works in world model or video generation literature (as cited in Line 171)? It would be valuable to include a controlled, head-to-head comparison, e.g., under fair conditions such as with or without camera parameters.
4. In Figure 5, why does adding packing improve overall performance (+Packing & Memory > +Memory)? My understanding is that packing should primarily improve efficiency, not necessarily task performance. Could the authors explain this effect?
5. Typo: Line 221: trajectory packing => Trajectory packing

---

> ### Author Response · Authors · 2025-11-21
>
> Thank you for your thoughtful comments.
>
> > W1, Q1
>
> Our technical contributions are threefold: the use of CDiT with RoPE (Section 4.1), Memory Retrieval (Section 4.2), and Trajectory Packing (Section 4.3). A clear drawback of the FramePack design is that frame importance is manually assigned rather than learned from data. To address this, we combine Memory Retrieval and Trajectory Packing to dynamically calculate which historical frames to attend to based on the agent's positional information, allowing for selective compression of the history.
>
> Figure 5 confirms that this mechanism achieves higher world modeling performance compared to designs that simply rely on manually assigning importance to recent frames. This advantage is further corroborated by results on a more realistic domain dataset [1] (please compare the "+ Packing" and "+ Packing & Memory" settings).
>
> | Nav. Range | Model | FVD (ABA) | FVD (ABCA) |
> |--|--|--|--|
> | 5 | NWM | **747** | 759 |
> | | WorldPack (ours) | 760 | **670** |
> | 15 | NWM | 665 | 773 |
> | | WorldPack (ours) | **551** | **669** |
> | 30 | NWM | 755 | 819 |
> | | WorldPack (ours) | **570** | **679** |
> | 50 | NWM | 841 | 810 |
> | | WorldPack (ours) | **562** | **455** |
>
> [1] Shah, et al. Rapid Exploration for Open-World Navigation with Latent Goal Models. CoLR2021.
>
> > Q2
>
> Regarding the statement that we "did not use camera parameters," we intended to convey that we did not employ Field of View (FOV)-based Monte Carlo estimation to calculate frame importance. Indeed, our proposed method calculates scores based solely on position and orientation (angle) information, without utilizing the camera's FOV.
> However, to prevent any potential misunderstanding, we have revised the corresponding section in the manuscript to clarify this point.
>
> > Q3
>
> In the original paper and additional experiments, we validated performance not only with the full WorldPack model but also with "Memory-only" and "Packing-only" configurations (please see W1). These experiments confirmed the superiority of our combined approach. In other words, we have already discussed and evaluated methods that, like ours, do not rely on FOV-based Monte Carlo estimation for frame importance.
>
> However, we agree that a direct comparison with existing methods such as WorldMem is valuable. While we have not yet completed this due to computational constraints, we are currently running these experiments and will add the results within the next week.
>
> > Q4
>
> The superiority of the "Packing + Memory" configuration over the "Memory-only" setting stems from its ability to condition on a significantly longer temporal context.
>
> Both the baseline and the "Memory-only" model are constrained to a token budget equivalent to 4 frames, effectively limiting their visibility to only 4 past frames (please refer to the "Context" and "Trajectory" columns). Specifically, the baseline utilizes the 4 most recent frames, whereas the "Memory-only" model utilizes 1 recent frame combined with 3 retrieved frames.
>
> In contrast, the introduction of Packing allows us to encode information from 19 frames into a token context equivalent to only 2.84 frames. In the "Packing + Memory" setting, this comprises 11 recent frames and 8 retrieved frames. Therefore, despite the reduced token count, the model gains access to information from a much larger number of frames. We attribute the performance improvement to this increased information density.
>
> | Method | Context Cost (Frame Equiv.) | Visible Trajectory Length | Frame Composition |
> | --- | --- | --- | --- |
> | Baseline (NWM) | 4.00 | 4 frames | 4 recent |
> | Memory-only | 4.00 | 4 frames | 1 recent + 3 retrieved |
> | Packing + Memory (Ours) | **2.84** | **19 frames** | 11 recent + 8 retrieved |
>
>
> > Q5
>
> Thank you for pointing out a typo. We have fixed our paper.

---

### Official Review · Reviewer_uLtW · 2025-11-01

**Soundness:** 3
**Presentation:** 3
**Contribution:** 2
**Rating:** 4
**Confidence:** 5

**Summary:**

The paper introduces WorldPack, a video world model that combines trajectory packing and memory retrieval to achieve long-horizon spatial consistency with short context lengths, building on a CDiT backbone with RoPE-based temporal embeddings to integrate memories from arbitrary distances. Trajectory packing hierarchically compresses past frames so that recent frames are high-resolution while older and retrieved memory frames are progressively downsampled, and the retrieval module selects context frames using a geometric score from agent positions and orientations when camera parameters are unavailable. Evaluation on the LoopNav benchmark in Minecraft shows consistent gains over previous methods across SSIM, LPIPS, PSNR, and DreamSim on both ABA and ABCA tasks with shorter contexts.

**Strengths:**

1. The proposed idea of combining trajectory packing with retrieval-based compression intuitively addresses the limitations of fixed-context world models.
2. Retrieval formulation does not rely on camera intrinsics and explicitly handles opposite-facing views at characteristic distances, improving practical applicability in game/sim settings.
3. Consistent gains on LoopNav across metrics and qualitative rollouts, with some ablations isolating the impact of retrieval vs packing and showing their complementarity.

**Weaknesses:**

1. My main concern is the latency/memory cost of the introduced retrieval-based compression approach, because it introduces many additional operations. Although claimed as “computationally efficient,” the paper does not analyze actual memory savings or inference latencies of the compressed memory versus previous baselines.
2. Evaluation is confined to a single simulator benchmark (Minecraft/LoopNav), and it is more important to test beyond simulators and towards real-world data to validate generality, like realestate10K​[1].
3. Improvements in metrics such as SSIM and PSNR are relatively modest, and perceptual evaluations rely heavily on qualitative claims. More statistical analysis, such as FID, FVD, or user studies, would lend stronger evidence.

[1] Zhou, Tinghui, et al. "Stereo magnification: Learning view synthesis using multiplane images." arXiv preprint arXiv:1805.09817 (2018).

**Questions:**

1. How does retrieval behave under significant localization noise or drift in agent pose, and can visual similarity cues be integrated when poses are partially wrong or missing?
2. What is the compute/time tradeoff of packing and retrieval at inference, and how does total context token count scale in practice across navigation ranges?
3. How far can the world model see? Or how many frames can the world model generate? When or at what number of frames do the generation results become blurry or unrealistic

I will raise my score if the author could address all of my concerns.

---

> ### Author Response · Authors · 2025-11-21
>
> Thank you for your insightful comments.
>
> > W1、Q2
>
> We present the single-step inference time and memory costs for the diffusion model. Compared to the baseline, WorldPack significantly extends the visible length of past trajectories from 4 to 19 frames. Although the incorporation of memory compression and retrieval processes introduces a slight overhead, the increase in inference time is marginal, at approximately 9%. Notably, memory consumption is actually reduced; this is because the compression mechanism lowers the number of tokens input into the CDiT (reducing the effective context from 4 frames to 2.84 frames, as shown in the context column). These experimental results corroborate WorldPack's ability to handle longer trajectory lengths with high computational efficiency.
> | Model | Context | Trajectory | Inference Time (1-step, sec) | Memory Usage (GB) |
> | --- | --- | --- | --- | --- |
> | Baseline | 4 | 4 | 0.430 | 22.08 |
> | WorldPack | 2.84 | 19 | 0.468 | 21.78 |
>
> > W2
>
> To verify the practical usefulness of WorldPack beyond simulator environments such as Minecraft, we conducted experiments using real-world data.
> Specifically, we evaluated our method on the RECON dataset [1], one of the commonly used datasets in prior video-generation world model studies [2, 3, 4].
> In our experiments, we used the first 20 frames as context and generated the subsequent frames.
> These results demonstrate that WorldPack achieves strong generative performance even on real-world data, confirming its effectiveness beyond simulated environments.
>
> | Model                   | Context | Trajectory | DreamSim ↓ | LPIPS ↓ | PSNR ↑ | SSIM ↑ |
> |-------------------------|---------|------------|-------------|---------|--------|---------|
> | NWM                     | 4       | 4          | 0.23        | 0.48    | 12.7   | 0.36    |
> | + Packing               | 2.84    | 19         | 0.18        | 0.45    | 13.4   | 0.40    |
> | + Packing & Memory      | 2.84    | 19         | **0.17**    | **0.44**| **13.6**| **0.40** |
>
> [1] Shah, et al. Rapid Exploration for Open-World Navigation with Latent Goal Models. CoLR2021.
> [2] Shah, et al. GNM: A General Navigation Model to Drive Any Robot. ICRA2023.
> [3] Sridhar, et al. Nomad: Goal masked diffusion policies for navigation and exploration. ICRA2024.
> [4] Bar, et al. Navigation World Models. CVPR2025.
>
> > W3
>
> We added the results of FVD to show stronger evidence.
> We confirmed that WorldPack outperformed the NWM baseline in the majority of settings, with the exception of short-term configurations.
> These results reinforce the evidence that WorldPack is highly effective for long-term video world modeling.
>
> | Nav. Range | Model | FVD (ABA) | FVD (ABCA) |
> |--|--|--|--|
> | 5 | NWM | **747** | 759 |
> | | WorldPack (ours) | 760 | **670** |
> | 15 | NWM | 665 | 773 |
> | | WorldPack (ours) | **551** | **669** |
> | 30 | NWM | 755 | 819 |
> | | WorldPack (ours) | **570** | **679** |
> | 50 | NWM | 841 | 810 |
> | | WorldPack (ours) | **562** | **455** |
>
> > Q1
>
> Although our current study did not explicitly address the impact of localization noise or drift, it is indeed possible to incorporate visual similarity cues, similar to the Memory Retrieval mechanism employed in WorldMem [5]. There is room for discussion regarding whether memory retrieval should rely primarily on localization features or visual features; this choice likely depends on the severity of the noise and drift in the localization data. Due to computational resource constraints, we are still working on these experiments and plan to provide the additional results within the coming week.
>
> [5] Xiao, et al. WORLDMEM: Long-term Consistent World Simulation with Memory. NeurIPS2025.
>
> > Q3
>
> Due to computational resource constraints, experiments involving the generation of longer video horizons are still in progress. We will provide the results within the coming week.

---

### Meta-Review · Area_Chair_awAt · 2026-01-10

**Summary:**

1. Novelty is incremental: trajectory packing mirrors FramePack; memory retrieval is a pose-scored bank; RoPE tweak is standard.

2. Only one simulator benchmark plus one small real-world set; no evidence on very long roll-outs (> 1 000 frames) or on other domains (driving, indoor-navigation).

3. Quantitative margins are modest (≈ 5–10 % on SSIM/LPIPS).

4. No user study or FID/FVD against methods beyond NWM baseline.

**Reviewer Concerns:**

1. Limited novelty / positioning

– All reviewers flagged that trajectory packing is essentially FramePack adapted to world-model data; memory retrieval and RoPE tweaks are known.

– Several asked for clearer differentiation from concurrent memory-based works (WorldMem, DrivingWorld, InfinityDrive).

2. Narrow empirical scope

– Original submission only evaluated on the Minecraft-derived LoopNav benchmark; no real-world sequences, no FVD, no comparison beyond NWM.

– Multiple reviewers (uLtW, wZPH, 42nA) explicitly demanded real-data experiments, longer roll-outs (> 1 000 frames), FVD curves, and released videos.

3. Modest metric gains

– Quantitative improvements (SSIM, LPIPS, PSNR, DreamSim) were small and statistically unverified; reviewers wanted user studies or FID/FVD.

4. Missing cost analysis

– Reviewers uLtW & wZPH required actual inference-time and memory tables to justify the “computationally efficient” claim.

5. Writing / presentation issues

– Typos, unclear camera-parameter assumptions, and the counter-intuitive title phrase “Compressed Memory Improves Spatial Consistency” were noted.

**Reviewer Scores:**

N.A.

---

### Decision · Program_Chairs · 2026-01-26

Reject